# RECOGNIZE ANY SURGICAL OBJECT: UNLEASHING THE POWER OF WEAKLY-SUPERVISED DATA

**Jiajie Li**[1]    **Brian R. Quaranto**[2]    **Chenhui Xu**[1]    **Ishan Mishra**[1,3]
**Ruiyang Qin**[1,4]    **Dancheng Liu**[1]    **Peter C. W. Kim**[2]*    **Jinjun Xiong**[1]*
[1]Department of Computer Science and Engineering, University at Buffalo
[2]Department of Surgery, University at Buffalo
[3]Department of Computer Science and Engineering, IIT, Jodhpur
[4]Department of Computer Science and Engineering, University of Notre Dame

## ABSTRACT

We present RASO, a foundation model designed to **R**ecognize **A**ny **S**urgical **O**bject, offering robust open-set recognition capabilities across a broad range of surgical procedures and object classes, in both surgical images and videos. RASO leverages a novel weakly-supervised learning framework that generates tag-image-text pairs automatically from large-scale unannotated surgical lecture videos, significantly reducing the need for manual annotations. Our scalable data generation pipeline gathers 2,200 surgical procedures and produces 3.6 million tag annotations across 2,066 unique surgical tags. Our experiments show that RASO achieves improvements of 2.9 mAP, 4.5 mAP, 10.6 mAP, and 7.2 mAP on four standard surgical benchmarks respectively in zero-shot settings, and surpasses state-of-the-art models in supervised surgical action recognition tasks.

**This paper includes surgical images that may be distressing to some readers.**

## 1 INTRODUCTION

Surgical object recognition and segmentation serve as two foundational pillars of computer vision in surgery, driving advancements that enable machines to accurately interpret and interact with complex surgical visual data by identifying and classifying surgical objects, as well as delineating their boundaries. Great efforts have been made in surgical segmentation, with models such as ISINet (González et al., 2020), Matis (Ayobi et al., 2023), and S3Net (Baby et al., 2023). However, due to the limitations of their training datasets, such as the small scale and limited annotation variety, they follow a closed-set design. For example, widely-used surgical segmentation datasets like EndoVis17 (Allan et al., 2019) and EndoVis18 (Allan et al., 2020) have only eleven categories of objects. These closed-set approaches stop these models from generalizing beyond seen classes during training, limiting their effectiveness in more diverse and dynamic real-world surgical environments.

Recent advances in general computer vision have introduced the Segment Anything Model (SAM) (Kirillov et al., 2023), which has significantly enhanced models' localization capabilities by enabling high-quality segmentation across a wide variety of objects, even on unseen categories. Building on SAM, models like SurgicalSAM (Yue et al., 2024) and MedSAM (Ma et al., 2024) have been adapted specifically for surgical images. These models leverage SAM's powerful localization abilities to achieve more accurate and efficient surgical object segmentation. However, similar to SAM, while they excel in localization, their recognition ability remains limited. For example, to obtain segmentation results using MedSAM, manual inputs such as bounding boxes need to be provided to guide the model. On the other hand, SurgicalSAM, though independent of manual inputs like bounding boxes, still requires the user to provide text prompts to guide the segmentation process. Additionally, SurgicalSAM is only fine-tuned on EndoVis17 and EndoVis18 datasets which cover eleven classes. Due to these limitations in recognition and the reliance on manual input, these models are unable to automatically generate segmentation mask-category pairs, which are essential for real-world applications like real-time robotic-assisted surgeries (Diana & Marescaux, 2015; Lanfranco et al., 2004). This restricts their utility in dynamic and diverse surgical environments where automation and adaptability are critical.

---

*For correspondence, contact `jinjun@buffalo.edu`, `pckim@buffalo.edu`

Surgical Object Recognition

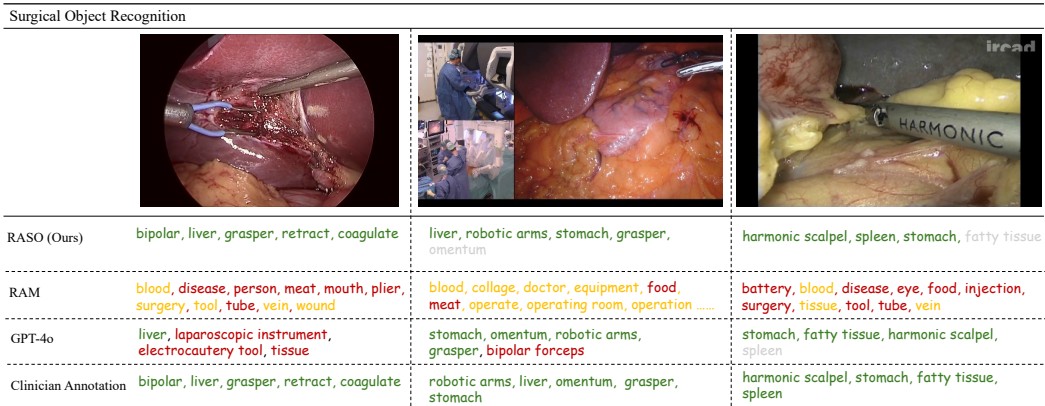

| | | | |
|---|---|---|---|
| RASO (Ours) | bipolar, liver, grasper, retract, coagulate | liver, robotic arms, stomach, grasper, omentum | harmonic scalpel, spleen, stomach, fatty tissue |
| RAM | blood, disease, person, meat, mouth, plier, surgery, tool, tube, vein, wound | blood, collage, doctor, equipment, food, meat, operate, operating room, operation ...... | battery, blood, disease, eye, food, injection, surgery, tissue, tool, tube, vein |
| GPT-4o | liver, laparoscopic instrument, electrocautery tool, tissue | stomach, omentum, robotic arms, grasper, bipolar forceps | stomach, fatty tissue, harmonic scalpel, spleen |
| Clinician Annotation | bipolar, liver, grasper, retract, coagulate | robotic arms, liver, omentum, grasper, stomach | harmonic scalpel, stomach, fatty tissue, spleen |

Figure 1: Comparison of surgical object recognition performance across different models and clinician annotations. Green for correct tags. Red for wrong tags. Yellow for tags are not accurate but somewhat related. Grey for tags are missing. RASO demonstrates high accuracy in identifying relevant surgical instruments and anatomical structures, aligning closely with clinician annotations.

To address the limited recognition abilities of models like SAM, recent advancements have introduced the Recognize Anything Model (RAM ; Zhang et al., 2024). RAM's broad recognition capabilities have enabled progress in grounded segmentation, where the model directly outputs label-segmentation pairs from a given image without any human inputs. Grounded SAM (Ren et al., 2024) combines RAM's recognition strengths with models like Grounding DINO (Liu et al., 2023), which generates location anchors for objects in an image and then applies SAM to produce precise segmentation masks. However, to achieve grounded segmentation in the surgical domain, a model with strong surgical object recognition capabilities is required.

Therefore, to bridge the gap between recognition and segmentation in the surgical domain, we introduce a foundation model designed to **R**ecognize **A**ny **S**urgical **O**bject (RASO). Developing such a model faces a key challenge that has also hindered progress in previous work: the scarcity of large-scale annotated surgical datasets, as manual annotation for surgical images requires specialized expertise and is highly time-consuming. To address this, we propose a data generation pipeline with minimal human intervention that generates a large-scale, weakly-supervised dataset for training such a model. Another challenge lies in the importance of video modality in the surgical domain, where accurately capturing surgical actions in surgical videos is crucial. While frame-by-frame recognition using image-based models is possible, it loses important temporal information. To address this, our work improves RAM by introducing an attention-based temporal-fusion mechanism that improves the ability to recognize surgical actions with faster inference speed.

Our major contributions to this work are as follows:

- *RASO.* We present a novel open-set recognition model in the surgical domain, capable of recognizing a wide range of surgical objects from both images and videos. RASO incorporates a temporal-attention fusion layer to effectively capture the temporal relationships between video frames, improving the ability to capture surgical actions. As shown in Fig. 1, RASO demonstrates surgical object recognition aligned with clinician expertise.
- *Weakly-supervised Learning Framework.* We present a novel weakly-supervised learning framework that leverages unannotated surgical data to generate tag-image-text pairs automatically. This approach significantly reduces reliance on manual annotations, making it broadly applicable to other vertical domains where annotated data is scarce. The cost-efficiency of our approach allows us to train RASO within 8 hours using 8 A6000 GPUs.
- *Weakly-supervised Data Generation Pipeline.* We introduce a scalable data generation pipeline that curates a comprehensive dataset from 2,200 unannotated surgical procedures, processing 901K images and 1.2M sentences. This pipeline generates 3.2M tag annotations from 2,066 unique surgical tags, all with minimal human intervention.
- *Comprehensive Experiments.* We conduct extensive experiments, where RASO achieves improvements of 2.9 mAP, 4.5 mAP, 10.6 mAP, and 7.2 mAP across four standard surgical datasets in zero-shot settings. In the supervised setting, RASO also surpasses state-of-the-art (SOTA) models on the surgical action triplet recognition task. We validate the effectiveness of each component in our method through ablation studies.

## 2    RELATED WORK

**Vision-Language Models.**   Recent advancements in vision-language models have significantly influenced multimodal tasks such as image captioning and cross-modal retrieval. Pioneering works such as CLIP (Radford et al., 2021) and ALIGN (Jia et al., 2021) introduced contrastive learning approaches, enabling zero-shot image classification and retrieval by aligning image and text representations. CLIP's ability to generalize across tasks has been instrumental, while ALIGN further scaled pretraining to billions of noisy image-text pairs. Florence (Yuan et al., 2021) and Flamingo (Alayrac et al., 2022) expanded on these ideas, incorporating task-specific fusion mechanisms and deeper multimodal integration for applications in VQA and image captioning. More recently, LLaVA (Liu et al., 2024) and BLIP (Li et al., 2022) explored the potential of large-scale vision-language pretraining in interactive and fine-tuned tasks, providing enhanced multimodal understanding. While these general-domain models have limited application in the surgical field, recent works such as LLaVA-Surg (Li et al., 2024), which focuses on surgical video question answering through the Surg-QA dataset, have extended vision-language models to address domain-specific challenges in surgery. SurgVLP (Yuan et al., 2023) leverages surgical video lectures and multimodal contrastive learning, yet suffers from potential performance degradation due to unregulated image-text pairs that may introduce noise, such as irrelevant textual information not aligned with the surgical content. Distinguished from existing works, our approach employs image-tag-text pairs for training, effectively regulating content and enhancing the quality of generated text based on assigned tags.

**Image Tagging** is the task of automatically assigning multiple descriptive labels to an image. Traditional methods utilize network architectures like ResNet (He et al., 2015) or Vision Transformers (Dosovitskiy, 2020), trained on large-scale manually annotated datasets such as PASCAL VOC (Everingham et al., 2015) and COCO (Lin et al., 2014), with Binary Cross-Entropy (BCE) loss for multi-label prediction. However, these approaches struggle to generalize beyond the labels present in these datasets. Recently, contrastive learning-based methods, such as CLIP (Radford et al., 2021), have demonstrated strong performance in aligning image and text embeddings. By utilizing a predefined tag set and setting a threshold, these models can also be applied for multi-label recognition. Tag2Text (Huang et al.) introduces image tagging into vision-language models by leveraging tags parsed from image-paired text to guide learning, while RAM (Zhang et al., 2024) builds on this approach by training a foundation model using large-scale annotation-free image-text pairs to achieve superior zero-shot image tagging performance. Building upon RAM, our work leverages large-scale image-text pair training to advance surgical object recognition.

**Surgical Image Recognition** involves using computer vision to automatically detect and classify surgical instruments, anatomical structures, actions, and phases in surgery videos or images , which can be utilized for skill assessment (Lam et al., 2022), surgical workflow understanding (Hu et al., 2025; Yuan et al., 2024), and robotic-assisted surgery (Diana & Marescaux, 2015; Rivero-Moreno et al., 2023). Datasets such as EndoVis17 and EndoVis18 provide annotations for instruments and organs, which are commonly used for training and evaluating models in tasks like tool detection and segmentation. Recent works have introduced action triplet annotations to further enhance surgical video understanding. Nwoye et al. (2020b) proposes the CholecT40 dataset annotates surgical images with action triplets, describing the instrument used, the action performed, and the target anatomy. Tripnet (Nwoye et al., 2020c) was later designed to recognize these triplets. With advancements in the attention mechanism, the Rendezvous model (Nwoye et al., 2022), which achieves SOTA recognition performance, was later proposed alongside an extended dataset, CholecT50. We evaluate RASO on the CholecT50 dataset in both zero-shot and supervised settings, comparing it with Rendezvous in the supervised scenario. In contrast to previous works, our model is more scalable and supports open-set recognition, allowing it to generalize to unseen classes during inference. Furthermore, our pretraining data is significantly larger, improving the robustness and transferability of our model to a wide range of surgical object recognition tasks.

## 3    RASO

### 3.1    ARCHITECTURE

The architecture of RASO is built on RAM (Zhang et al., 2024). As shown in Fig. 2, RASO consists of four key components: an image encoder, a spatial-temporal fusion layer, a tag decoder for surgical

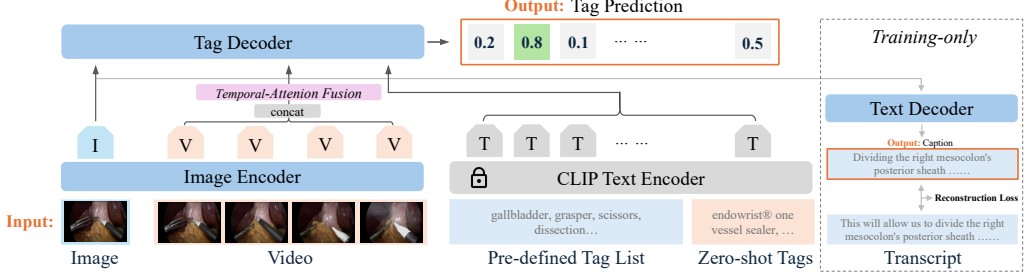

Figure 2: RASO Architecture. RASO includes (1) an image encoder to extract visual features from images and video frames; (2) a temporal-attention fusion layer to handle video inputs by capturing temporal dependencies across frames; (3) a tag decoder to predict surgical labels by combining visual and tag embeddings; and (4) a text decoder to align the visual content with transcriptions.

label prediction, and a text decoder for aligning visual and textual representations during pretraining. The core improvement over RAM is the introduction of a temporal fusion layer for efficient video recognition.

**Image Inference.** For image inputs, we utilize the Swin-Transformer (Liu et al., 2021) to project the images to image features. We leverage the CLIP text encoder to embed the predefined tag list. The tag decoder then combines the visual features and the corresponding tag embeddings to predict logits for each tag. Finally, a predefined threshold is applied to identify the recognized objects.

**Video Inference & Temporal-Attention Fusion Layer.** For video inputs, we uniformly sample $N$ frames at regular intervals and compute image embeddings using the same image encoder as for image inputs. The encoder generates frame-level features, denoted as $h_{\text{image}} \in \mathbb{R}^{T \times D}$, where $T$ represents the number of image tokens and $D$ is the feature dimension. These individual frame features are stacked along the temporal axis, resulting in a combined tensor $h_{\text{images}} \in \mathbb{R}^{N \times T \times D}$, capturing both spatial information and temporal progression. Afterward, the temporal-attention fusion layer is applied to aggregate temporal information across frames. Using a multi-head attention mechanism, the layer captures temporal dependencies between frames by reshaping and permuting the input into a format suitable for attention computation. The resulting tensor, $h_{\text{video}} \in \mathbb{R}^{T \times D}$, is seamlessly decoded using the same tag decoder as for image inputs, ensuring consistent prediction across both modalities. As shown in Section 5.3, the temporal-attention fusion effectively captures dynamic information, particularly for categories that require temporal context to be correctly identified, such as surgical actions, thus enhancing video recognition performance.

**Training.** During the training stage, a text decoder reconstructs the image caption or transcriptions from surgical lectures based on the visual features and predicted tags. This strengthens the alignment between visual content and textual information, improving the model's ability to connect visual representations with semantic meaning. The text decoder is not used during inference.

**Open-Vocabulary Inference.** For unseen tags, the model uses the CLIP text encoder to generate embeddings, updating the tag embedding list accordingly. By leveraging the semantic relationships between known and unseen tags, the model can effectively generalize to new tags.

## 3.2 TRAINING

The training process involves two stages: pretraining and fine-tuning.

**Stage 1: Pretraining.** In this stage, we pretrain the model using 901K weakly supervised image-tag-text triplets extracted from WebSurg surgical lectures. The core objective during pretraining is to align the visual features extracted from surgical videos with the corresponding textual information. We keep the CLIP text encoder frozen and fine-tune the rest of the model for 10 epochs.

**Stage 2: Fine-Tuning.** In the fine-tuning phase, we fine-tune RASO with a mixture of image-tag and video-tag pairs. This combination enriches the model's ability to generalize across diverse data formats. For zero-shot experiments, we exclusively relied on VLM-generated data as discussed in

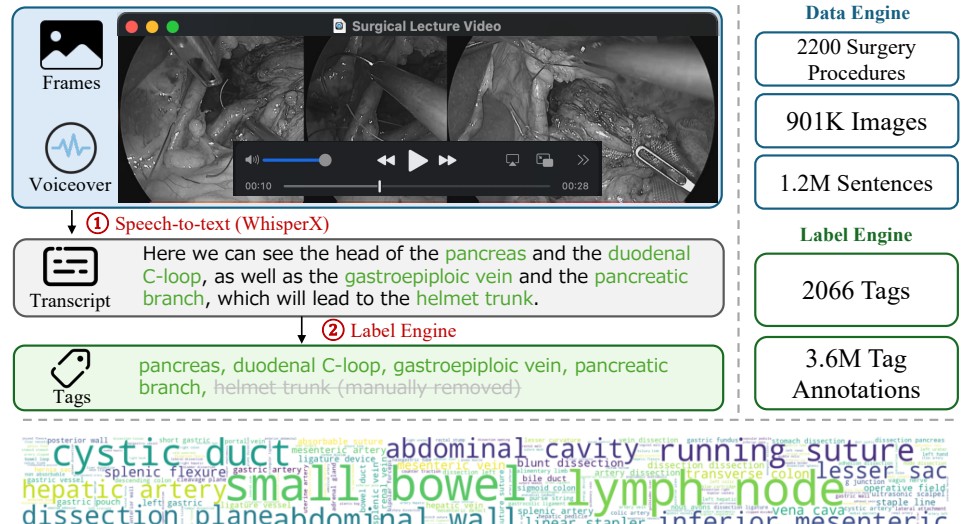

Figure 3: The weakly supervised pretraining data pipeline (top-left). Surgical lecture videos, including both visual frames and voiceover transcripts, are processed through named entity recognition, semantic parsing, and manual filtering to extract relevant surgical tags. These tags encompass anatomical structures, instruments, and procedural information. The bottom figure gives an overview of the tag frequency in our pretraining dataset using a word cloud.

section 4 with additional image captioning data. We keep the CLIP text encoder frozen, fine-tune the rest of the model for 4 epochs.

**Implications.** Our approach offers two major advantages: *(i) Cost Efficiency.* We pretrained the model in less than 5 hours using 8 NVIDIA A6000 GPUs to process 901K samples over 10 epochs. Fine-tuning took 3 hours on the same hardware, demonstrating the scalability and resource efficiency of our method. *(ii) Broad Applicability.* The method is applicable across other vertical domains where annotated data is scarce, but videos with voiceovers are abundant. For instance, medical instructional videos, which often include voiceovers, are ideal candidates for weakly supervised learning. This adaptability makes our method suitable for various fields with similar data structures.

## 4 WEAKLY-SUPERVISED DATA GENERATION

To pretrain a robust foundation model for surgical object recognition, we curated a large-scale dataset capturing the complexity and diversity of surgical environments. As shown in Fig. 3, the dataset construction involved two main components: a *data engine* and a *label engine*. The label engine extracts relevant surgical tags from textual data, while the data engine aligns these tags with visual data from surgical lecture videos.

### 4.1 DATA ENGINE

The data engine was developed to collect large-scale visual and textual data from surgical lecture videos. Specifically, we sourced 2,200 surgical procedures from platforms like WebSurg (WebSurg, 2024), generating 901K images and 1.2 million sentences through WhisperX (Bain et al., 2023), which produced time-aligned transcriptions for each video. These videos were segmented into clips corresponding to individual sentences, and non-visual content, such as lecture slides, was filtered out using *GPT-3.5-Turbo* (Brown, 2020). The filtered video clips, along with their transcriptions, were processed by the label engine as described in Section 4.2 to generate weakly supervised tag annotations. In total, through the integration of the data and label engines, we constructed a visual surgical vocabulary consisting of 2,066 unique tags—1,455 for pretraining and 611 for fine-tuning—and produced 3.6 million tag annotations, with 3.2 million used for pretraining and 0.4 million for fine-tuning. Frames were sampled from these annotated clips, with each frame paired alongside the corresponding transcription and tags. These image-text-tag triplets served as the foundation for RASO's pretraining and fine-tuning stages, as outlined in Section 3.2.

## 4.2 LABEL ENGINE

Our goal was to build a comprehensive set of surgical tags encompassing organs, instruments, surgical actions, procedures, and phases. These tags form the model's vocabulary, enabling it to recognize a wide variety of elements within surgical scenes. Our label engine consists of three parts:

**Surgical Entity Recognition.** To curate a comprehensive set of static, domain-specific tags for surgical object recognition, we utilized spaCy's biomedical model, `en_core_sci_md` (Neumann et al., 2019), to extract key entities such as anatomical structures, surgical instruments, and procedural terms from surgical lecture transcripts. This model, trained extensively on medical literature, excels in identifying Named Entities (NER) that are essential for understanding surgical contexts. By processing over 1.2 million sentences, we generated a wide-ranging set of relevant terms, capturing static objects like organs, tissues, and surgical tools, including scalpels and graspers. This process of entity recognition served as the backbone for developing a robust vocabulary, enabling the model to effectively recognize and differentiate critical elements within complex surgical scenes.

**Surgical Action Extraction.** To capture the dynamic processes within surgical procedures, we employed SceneGraphParser (Wu et al., 2019) to extract detailed verb-noun pairs from surgical lecture transcripts. This parser identifies verb phrases representing actions and links them to their corresponding objects, such as surgical tools, anatomical structures, or tissues. Through this process, we automatically generate structured action triplets, such as <grasper dissects gallbladder>, where the verb "dissects" represents the action, "grasper" is the instrument acting, and "gallbladder" is the object being operated on. By extracting these action-object relationships, the model can understand both the activity taking place and the entities involved. This approach enriches the dataset with contextual information about the interactions between surgical tools and anatomical structures, which is crucial for accurately modeling the procedural flow and dynamics of surgical tasks.

**VLM Automated Annotation.** Recent studies (Li et al., 2023; Cheng et al., 2023; Hou & Ji, 2024) have shown that models like GPT-4o can effectively recognize biomedical and surgical images. Leveraging this capability, we used GPT-4o to annotate 120K surgical images, generating 372K tag annotations with 120K image captions. These auto-generated annotations enriched our dataset, providing additional weak supervision and reducing the need for manual labeling. A detailed evaluation of GPT-4o-generated annotations is provided in Appendix A.1.

## 5 EXPERIMENTS

In this section, we present the experimental setup and results of RASO for various surgical object recognition tasks for both image and video inputs. We start by detailing the benchmarks, datasets, and evaluation metrics used to assess the model's performance. Next, we address the following questions: *(i)* Can RASO successfully perform zero-shot surgical object recognition without manual annotations? *(ii)* How does RASO compare to SOTA models in supervised recognition tasks? *(iii)* Does the temporal fusion layer improve accuracy and efficiency in video recognition tasks? Lastly, we conduct ablation studies to explore the contribution of each component in our method.

## 5.1 EXPERIMENT SETTING

**Benchmarks.** We evaluated the RASO model on several well-established surgical datasets. The GraSP (Ayobi et al., 2024) dataset supports multi-level surgical scene understanding, with tasks spanning from phase and step recognition to fine-grained instrument segmentation and visual action detection. CholecT50 (Nwoye et al., 2022) focuses on recognition tasks, offering annotations in the form of action triplets (<*instrument, verb, target*>), phase labels, and bounding boxes, making it suitable for both supervised and zero-shot evaluations. Cholec80 (Twinanda et al., 2016) provides 80 annotated laparoscopic cholecystectomy videos, focusing on phase recognition and tool detection. EndoVis18 (Allan et al., 2020) is centered around robotic-assisted surgery, requiring the segmentation of anatomical structures and medical devices. We specifically evaluate on the Robotic Scene Segmentation (RSS) subset of Endovis18. GraSP, Cholec80, and RSS are segmentation-focused datasets, for which we generated tag annotations by leveraging the presence of segmentation masks in their annotations. We provide the details of these benchmarks in Table 1.

**Evaluaiton Metrics.** To evaluate the performance of the RASO model, we employ four widely used metrics in object recognition tasks: Precision (P), Recall (R), F-beta score ($F_\beta$) and mean Average Precision (mAP). When evaluating Precision and Recall metrics, we apply the same rule to find the best threshold for each method which maximizes the $F_{\beta=0.5}$ score, we choose $F_{\beta=0.5}$ to balance the precision and recall. When evaluating on CholecT50, we use the same method as Rendezvous (Nwoye et al., 2022) which utilizes the *ivtmetrics* (Nwoye & Padoy, 2022) library.

**Baselines.** *(i) Zero-shot Recognition.* We compare the RASO model with several contrastive learning-based approaches, including CLIP (Radford et al., 2021), BiomedCLIP (Zhang et al., 2023), PubMedCLIP (Eslami et al., 2023), and SurgVLP (Yuan et al., 2023), to evaluate mAP based on the similarity between input images and labels. CLIP, a general-domain model trained on large-scale image-text pairs, serves as a baseline for non-domain-specific tasks. BiomedCLIP, designed for the biomedical domain, achieves SOTA performance on image-text tasks using data from scientific articles, while PubMedCLIP fine-tunes CLIP for medical visual question-answering tasks using PubMed articles. SurgVLP, specifically developed for the surgical domain, leverages surgical video lectures and multi-modal learning to perform zero-shot recognition of surgical objects without requiring manual annotations, demonstrating strong adaptability across surgical tasks. *(ii) Supervised Recognition.* We compare RASO with Tripnet (Nwoye et al., 2020c), Attention Tripnet (Nwoye et al., 2022), and the SOTA Rendezvous (Nwoye et al., 2022) on the CholecT50 dataset for surgical action triplet detection. Tripnet first introduces a basic framework for recognizing surgical action triplets, focusing on mapping the relationships between instruments, actions, and targets. Attention Tripnet enhances this by adding attention mechanisms, improving recognition accuracy by focusing on key elements in the frames. Rendezvous further advances the model with spatial attention to target specific regions and semantic attention to capture the intricate relationships between triplet components, leading to superior performance in recognizing surgical interactions.

**Implementation.** *(i) Data Generation.* We utilized the "large-v2" version of WhisperX for generating transcriptions, followed by data filtering with "gpt-3.5-turbo-0125". For additional annotation, we employed "gpt-4o". *(ii) Training Data.* For both zero-shot and supervised tasks, we use the self-generated weakly supervised data for pretraining. For zero-shot tasks, we finetune RASO with GPT-4o annotated data. For the supervised task, we finetune the model with the training split of the CholecT50 dataset. *(iii) RASO Model.* We initialize the image encoder using "swin-large" weights of the Swin-Transformer, with an input image size of 384×384. We used the text encoder of CLIP version "ViT-B/16" for tag embeddings. *(iv) Training Details.* We used the AdamW (Loshchilov, 2017) optimizer for both pretraining and fine-tuning in our zero-shot and supervised experiments. During pretraining, the weight decay was set to 0.05, with an initial learning rate of 1e-4, a minimum learning rate of 5e-7, and a learning rate decay rate of 0.9. The warmup learning rate was 5e-7, and the warmup steps were set to 3000. Pretraining was conducted for a maximum of 10 epochs with a batch size of 26 per device. For fine-tuning, the weight decay remained at 0.05, the initial learning rate was set to 5e-6, and the minimum learning rate was 0. The fine-tuning process lasted for 4 epochs, with a batch size of 26 per device. *(v) Hardware.* We train all the models on 8 × NVIDIA A6000 GPUs. We evaluate the latency on one NVIDIA A6000 GPU.

## 5.2 RESULTS

**Zero-shot Image Recognition.** We evaluate the zero-shot recognition capabilities of RASO and compare it against several competitive baselines across multiple surgical datasets, including CholecT50, Cholec80, RSS, and GraSP. The results, as summarized in Table 2, highlight the effectiveness of RASO in recognizing surgical objects without the need for explicit annotations.

As shown in Table 2, RASO consistently outperforms other models, particularly in challenging scenarios such as CholecT50 and GraSP, where domain-specific knowledge plays a crucial role in

Table 1: Details of test benchmarks.

| Dataset | # Category | # Image |
|---|---|---|
| CholecT50 | 29 | 19,205 |
| Cholec80 | 7 | 98,194 |
| GraSP | 29 | 2,861 |
| RSS (Endovis18) | 11 | 997 |

Table 2: Zero-shot Surgical Image Object Recognition. "FT" is fine-tuning. $\beta = 0.5$. We use **bold** to indicate the best result in a column, and underline to indicate the second best.

| Methods | CholecT50 | | | | Cholec80 | | | | RSS | | | | GraSP | | | |
|---|---|---|---|---|---|---|---|---|---|---|---|---|---|---|---|---|
| | P | R | $F_\beta$ | mAP | P | R | $F_\beta$ | mAP | P | R | $F_\beta$ | mAP | P | R | $F_\beta$ | mAP |
| CLIP | 16.6 | 64.7 | 17.9 | 15.4 | 20.6 | 52.2 | 20.0 | 17.4 | 57.2 | 42.3 | 53.4 | 42.2 | 13.2 | 64.9 | 14.6 | 10.8 |
| BiomedCLIP | 18.9 | 65.6 | 19.7 | 15.5 | 28.0 | 72.2 | 27.1 | 17.6 | 50.6 | 70.1 | 51.2 | 42.5 | 21.6 | 46.2 | 21.7 | 11.6 |
| PubMedCLIP | 16.9 | 57.9 | 18.2 | 15.4 | 20.7 | 70.1 | 21.4 | 17.4 | 48.0 | 74.7 | 48.7 | 42.1 | 13.4 | 76.1 | 15.3 | 10.8 |
| SurgVLP | 22.6 | 39.6 | 22.6 | 19.7 | 33.9 | 27.3 | 31.1 | 33.3 | 52.8 | 60.6 | 50.8 | 48.2 | 19.9 | 38.7 | 19.0 | 12.8 |
| RASO (w/o FT) | 22.8 | 48.8 | 23.0 | 20.6 | 26.0 | 50.1 | 27.6 | 25.0 | 60.9 | 70.8 | 58.0 | 53.3 | 23.5 | 50.3 | 20.4 | 16.4 |
| RASO | 27.0 | 43.3 | **25.2** | **22.6** | 40.7 | 38.3 | **39.2** | **37.8** | 66.1 | 58.2 | **65.3** | **58.8** | 25.2 | 52.0 | **24.1** | **20.0** |

Table 3: Comparison with SOTA Models on CholecT50 Dataset (mAP).

| Methods | # Supported Tags | CholecT50 | | | |
|---|---|---|---|---|---|
| | | Instrument (6) | Verb (9) | Target (14) | All (29) |
| Tripnet (Nwoye et al., 2020a) | 29 | **92.1** | 54.5 | 33.2 | 52.0 |
| Attention Tripnet (Nwoye et al., 2022) | 29 | 92.0 | 60.2 | 38.5 | 56.3 |
| Rendezvous (Nwoye et al., 2022) | 29 | 92.0 | 60.7 | 38.3 | 56.4 |
| RASO (w/o pretraining) | open-set | 83.3 | 59.0 | 40.0 | 54.8 |
| RASO | open-set | 88.3 | **63.4** | **40.5** | **57.5** |

object recognition. On CholecT50, RASO achieves the highest mAP score (22.6%) and $F_{\beta=0.5}$ score (25.2), showcasing its ability to detect fine-grained surgical action triplets effectively. For Cholec80, RASO surpasses other methods with a significant improvement in Precision (40.7%) and an mAP of 39.2%, indicating strong zero-shot capabilities in both instrument detection and phase recognition tasks. RSS results further highlight the robustness of RASO, as it achieves the best performance in both $F_{\beta=0.5}$ (58.2%) and mAP (65.3%), demonstrating its superior ability to generalize to unseen tasks. In comparison, while models such as CLIP and BiomedCLIP perform well in general image-text tasks, they fall short in domain-specific tasks like surgical object recognition, where RASO's fine-tuning on weakly-supervised surgical datasets provides a distinct advantage. Finally, in GraSP, a dataset emphasizing complex visual action detection, RASO again demonstrates superior performance, achieving a notable mAP of 20.0%. This highlights its adaptability across a variety of surgical environments, including multi-task datasets requiring detailed visual understanding.

Notably, even without fine-tuning with GPT-4o annotations, RASO (w/o FT) outperforms all other approaches across datasets, except for Cholec80, where it is marginally outperformed by SurgVLP.

**Supervised Image Recognition.** For supervised evaluation, we compare our model, RASO, with several baselines on the CholecT50 dataset, which provides detailed annotations for fine-grained triplet recognition tasks. The baselines include Tripnet, Attention Tripnet, and Rendezvous proposed by Nwoye et al. (2022), which aim to recognize *<instrument, verb, target>* triplets from surgical videos. Our proposed RASO model, significantly outperforms these baselines, especially in the verb and target categories. As shown in Table 3, RASO achieves a mean score of 57.5, an improvement over the baseline models in both fine-grained recognition and task transfer. Through pretraining, RASO achieves superior performance in verb classification and overall performance, surpassing the SOTA Rendezvous model. This demonstrates the effectiveness of our pretraining strategy. Notably, our method is only fine-tuned on the CholecT50 dataset for 4 epochs, while Rendezvous is trained for over 200 epochs. Therefore, we gain much less performance from data augmentation compared to Rendezvous. Despite this, RASO still achieves superior results.

**Supervised Video Recognition.** We compare two approaches for surgical video object recognition: RASO (video) and RASO (image-based). RASO (video) is the default approach that employs the temporal fusion mechanism that processes sequences of video frames by aggregating information across time, capturing temporal dependencies for more context-aware recognition as described in Section 3.1. This method is optimized for direct video input, leading to faster inference times and higher accuracy. In contrast, RASO (image-based)

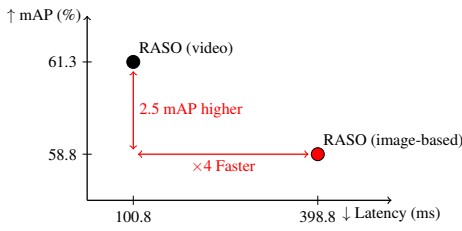

Figure 4: Comparison on Video Inference.

Table 4: Supervised video recognition on CholecT50 (mAP).

| Pretraining | Temporal Fusion | Average Pooling | Instrument | Verb | Target | All |
|:---:|:---:|:---:|:---:|:---:|:---:|:---:|
| ✓ | ✓ | | 88.9 | 74.5 | 41.0 | 61.3 |
| ✓ | | ✓ | 90.3 | 68.5 | 41.7 | 60.1 |
| | ✓ | | 84.7 | 67.6 | 39.2 | 57.4 |

leverages the image inference pipeline for video input which processes each video by running inference independently for each frame in a batch and then combines the detected tags using a union operation to produce the final recognition result for the video. While this approach simplifies the pipeline, it lacks the ability to leverage temporal information, limiting its performance in tasks where frame-to-frame context is essential. As shown in Fig. 4, RASO (video) achieves a higher mAP of 61.3%, outperforming the RASO (image-based) method by 2.5%, with an mAP of 58.8%. Additionally, RASO (video) is about $4\times$ faster, with a latency of 100.8 ms compared to 398.8 ms for the image-based model. Note that, the slight mAP difference is partly due to the label merging process in the image-based approach, which can introduce inconsistencies when aggregating tags from individual frames.

## 5.3 ABLATION STUDY

To analyze the contributions of various components in the RASO model, we conduct ablation studies for both zero-shot and supervised recognition tasks across different surgical datasets.

**Zero-shot Recognition.** We study the effectiveness of the fine-tuning stage which leverages GPT-4o-generated annotations to enhance performance. Table 2 shows the ablation study for supervised recognition on the CholecT50 dataset. Here, we observe that pretraining alone leads to strong performance, especially for instrument recognition, achieving 88.3%. As mentioned earlier, even without the fine-tuning stage, RASO (w/o FT) outperforms previous methods on three benchmarks. Adding GPT-4o data provides a further improvement, particularly for more complex categories like verbs and targets, underscoring the importance of GPT-4o in supervised learning as well.

**Supervised Recognition.** Table 3 shows that adding the pretraining stage significantly boosts performance. The model with pretraining achieves higher mAP across all categories, with a notable 57.5 overall mAP, surpassing the baseline's 54.8. This demonstrates the clear advantage of pretraining in improving recognition accuracy, especially for instruments and verbs.

**Video Recognition.** Table 4 compares the proposed temporal fusion layer against the naive average pooling layer for video recognition on the CholecT50 dataset. The results validate our hypothesis that the temporal fusion layer significantly improves verb (surgical action) recognition (74.5% vs. 68.5%), highlighting its advantage in capturing dynamic information critical for identifying actions. This also results in a higher overall mAP (61.3% vs. 60.1%), further demonstrating the effectiveness of leveraging temporal dependencies in surgical video analysis. Furthermore, incorporating pretraining significantly boosts performance across all metrics, achieving 5.2% improvement for instruments, 6.9% for verbs, and 1.8% for targets, with an overall mAP of 3.9%.

## 6 CONCLUSION

This paper introduces RASO, a foundation model for surgical object recognition. RASO marks a significant step forward in the surgical imaging domain. We showcase RASO's strong performance across zero-shot and supervised tasks and highlight the potential of scalable data generation techniques to support future advancements in surgical AI. RASO lays the groundwork for future advancements in grounded surgical segmentation, where recognition and segmentation tasks are intertwined. By providing accurate object recognition as a foundation, RASO has the potential to enhance segmentation models, enabling more precise and context-aware surgical scene understanding. By open-sourcing our code, model, and dataset, we aim to drive further research, bridging the gap between recognition and segmentation in surgical imaging applications.

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

# A APPENDIX

## A.1 CLINICIAN EXPERTS EVALUATION OF VLM AUTOMATED ANNOTATION.

Our motivation for leveraging GPT-4o in the annotation pipeline stems from its observed tendency to use common tags with reasonable accuracy. This property complements the broader tag coverage the pretraining dataset provides, creating a synergy that enhances model performance, especially on frequently occurring tags. To conduct an evaluation to assess the quality of annotations generated by GPT-4o, we randomly sampled 1,000 annotated images and assigned them to seven clinicians for expert tag annotations. Each clinician independently annotated the images with the most appropriate tags, which were then compared to the GPT-4o-generated annotations. We calculated the agreement rate, defined as the percentage of tags matched between GPT-4o annotations and clinician annotations. The evaluation shows results for 20 frequent tags, as outlined below:

Table 5: Agreement rates for GPT-4o annotations across 20 frequent tags.

| Tag | Agreement Rate |
|---|---|
| mediastinum | 0.35 |
| spleen | 0.32 |
| peritoneum | 0.32 |
| abdominal wall | 0.31 |
| stapler | 0.29 |
| intestine | 0.29 |
| pelvic cavity | 0.27 |
| needle driver | 0.27 |
| ligasure | 0.25 |
| lung | 0.25 |
| gallbladder | 0.24 |
| scissors | 0.22 |
| harmonic scalpel | 0.21 |
| trocars | 0.21 |
| hernia sac | 0.21 |
| choledochal cyst | 0.20 |
| adipose tissue | 0.20 |
| suture | 0.19 |
| laparoscopic grasper | 0.19 |
| skin | 0.19 |

