# OpenReview forum: "Recognize Any Surgical Object: Unleashing the Power of Weakly-Supervised Data"
_ICLR.cc/2025/Conference — ICLR 2025 Spotlight_

### Official Review · Reviewer_9rAR · 2024-10-31

**Soundness:** 3
**Presentation:** 3
**Contribution:** 2
**Rating:** 6
**Confidence:** 4

**Summary:**

The paper’s contributions include:

(1)RASO Model: A novel open-set recognition model tailored for recognizing diverse surgical objects in images and videos, incorporating temporal-attention fusion for video context.

(2)Weakly-supervised Learning Framework: An efficient framework leveraging unannotated data, significantly reducing annotation reliance and supporting domains with limited annotated data.

(3)Data Generation Pipeline: A scalable, weakly-supervised pipeline that generates extensive tag-image-text data from over 2,200 surgical procedures, producing comprehensive surgical annotations.

(4)Experimental Results: RASO demonstrates substantial mAP improvements in zero-shot and supervised settings, outperforming current state-of-the-art models in surgical action recognition.

**Strengths:**

(1)The development of RASO introduces an attention-based temporal-fusion mechanism specifically for surgical videos, where temporal information is essential but often neglected in frame-based models. Additionally, the weakly-supervised learning framework and data generation pipeline are highly creative solutions to the prevalent problem of data scarcity, enabling the model to learn from unannotated surgical videos at scale.

(2)Extensive testing on four benchmark datasets, both in zero-shot and supervised settings, provides strong evidence of the model's effectiveness.

(3)The article is well-structured, the motivation is clear, and the technical explanations are easy to understand

**Weaknesses:**

(1)The model heavily depends on a weakly-supervised dataset generated from surgical lecture videos, which, despite its scale, may introduce considerable noise due to the lack of domain-specific annotation quality control. If we could analyze the approximate distribution of entity keywords(tools or actions, etc.) in the captions, it would allow for a preliminary observation.

(2)While the temporal-attention fusion layer is a key innovation intended to enhance video-based recognition, the paper lacks a rigorous justification for why this layer outperforms simpler alternatives, such as average pooling or recurrent networks.

(3This architecture includes a label decoder and a text decoder, where the text decoder aligns the visual content with the transcript during training. However, is there potential redundancy between the label decoder and text decoder modules? Would an ablation study be necessary to verify this?)

**Questions:**

I'd like to ask how the frames were extracted. Were they keyframes, sampled at regular intervals, or something else?

---

> ### Author Response · Authors · 2024-11-19
>
> Thank you for your thoughtful review and insightful comments. We provide detailed responses to your questions and concerns below.
> > I'd like to ask how the frames were extracted. Were they keyframes, sampled at regular intervals, or something else?
>
> **R**:  We extract frames by sampling them at regular intervals from the videos. Specifically, for video recognition tasks, we uniformly sample a fixed number of frames (e.g., 15) at equal time intervals to ensure consistent temporal coverage across the video.
>
> > The model heavily depends on a weakly-supervised dataset generated from surgical lecture videos, which, despite its scale, may introduce considerable noise due to the lack of domain-specific annotation quality control. If we could analyze the approximate distribution of entity keywords(tools or actions, etc.) in the captions, it would allow for a preliminary observation.
>
> **R**: Thank you for pointing this out. To provide a preliminary observation of the entity keyword distribution, we have added a word cloud in Figure 3 (in the revised version) that gives an overview of the tag frequency in our dataset. In addition, to reduce noise in our weakly-supervised dataset, our method employs a domain-specific named entity recognition (NER) model to filter medical-related terms from the captions. Additionally, clinician experts were consulted to refine and validate high-frequency tags to ensure their relevance to surgical procedures. This multi-layered quality control approach helps maintain a high level of accuracy and domain-specific relevance in the dataset.
>
> > While the temporal-attention fusion layer is a key innovation intended to enhance video-based recognition, the paper lacks a rigorous justification for why this layer outperforms simpler alternatives, such as average pooling or recurrent networks.
>
> **R**: The temporal-attention fusion layer is designed to capture temporal dependencies between frames, which are critical for recognizing dynamic surgical actions. We also experimentally validated the effectiveness of the temporal fusion layer, as demonstrated in Table 4. For instance, temporal attention achieves a significant improvement of 6.0 mAP in action recognition tasks over the average pooling baseline, highlighting its effectiveness for capturing temporal information.
>
> > This architecture includes a label decoder and a text decoder, where the text decoder aligns the visual content with the transcript during training. However, is there potential redundancy between the label decoder and text decoder modules? Would an ablation study be necessary to verify this?
>
> **R**: The text decoder is implemented for text reconstruction following the setup in the original RAM paper. Its primary purpose is to provide additional textual semantic information during training, rather than serving as a core contribution of RASO. Given its auxiliary role, we did not conduct an ablation study specifically on this module.

---

> > ### Comment · Reviewer_9rAR · 2024-11-25
> >
> > Thank you to the authors for the clarification, and I will raise my rating. However, I hope the authors can reference some related works in the final manuscript, such as studies on other surgeries like ophthalmic surgeries: OphNet: A Large-Scale Video Benchmark for Ophthalmic Surgical Workflow Understanding.

---

> > > ### Author Response · Authors · 2024-11-26
> > >
> > > Thank you for recognizing our work and suggestions. In the latest revision, we have enriched the Related Work section to include studies on broader surgery types, such as OphNet.

---

### Official Review · Reviewer_A4T2 · 2024-11-01

**Soundness:** 3
**Presentation:** 3
**Contribution:** 3
**Rating:** 8
**Confidence:** 5

**Summary:**

The Paper presents a data generation pipeline for weakly supervised training framework and a foundation model architecture RASO. RASO predicts an open set of tags for single frames while taking video context into account. The data generation pipeline extracts relevant tags from voiceovers of surgical lecture videos. RASO is evaluated in a zero-shot and a supervised setting.

**Strengths:**

- interesting approach which proposes novel architecture, pipeline and data
- Paper is well structured, easy to read and easy to follow the argumentation
- profound evaluation, RASO is compared against 4 SOTA architectures for the zero-shot setting on 4 datasets and 3 SOTA models for the supervised setting on the CholecT50
- Paper presents an ablation study for the pretraining and fine-tuning stages, as well as two different aggregation methods for the video context understanding
- code, models and dataset will be open-sourced

**Weaknesses:**

- missing details regarding the evaluation, e.g. why was the video aggregation not used in the first two experiments? which data was used for which configuration? ... (see questions)
- no link to code, data, and models in the paper

**Questions:**

- Line 33: Two Datasets are called EndoVis17 and EndoVis18. In both years EndoVis consisted of 4 challenges releasing datasets. I propose to use the challenge names robotic instrument segmentation (RIS) & robotic scene segmentation (RSS)
- Figure 1 is not referenced in the text.
- Every paragraph cites methods or models when firstly used, even if previously already used
- Line 184ff: N and T both represent a number of features, the final embedding size is NxTxD, its unclear what the difference between N&T is and why there are two numbers of frames.
- Line 197 describes the text decoder to take visual features and predicted tags as input while figure 2 shows the encoder to take visual features and encoded tags but no prediction. Fig2 should be improved such that the prediction is taken into account by the text decoder.
- Line 235: The Paragraph heading “Discussion” doesn't really fit, as no results are discussed but rather advantages of the pipeline are listed
- Line 260: How are the generated tags split into the pretraining and fin-tuning set?
- Table1: EndoVis18 (RSS) is listed with 997 images. The dataset consists of 19 snippets with 300 frames each, why/how was it down sampled?
- Table2: why aren't the values of P & R highlighted? Esp. for P the text references the models superiority, this should be visualized in the table.
- Why is the Fbeta with beta=0.5 instead of the more common F1 score used?
- Table3: all values in the table should be given with the same amount of decimals
- Line 417ff: Figure2 introduces the RASO architecture to include a temporal-attention fusion, but the video recognition paragraph compares the video aggregation against a frame-wise approach? How does the frame-wise method work? Which features are aggregated?
- From the video recognition paragraph an Table3c it seems that the temporal-attention fusion was not used in the experiments of Table 1 and 2, as both configuration in Table3c differ from the earlier presented results (Table 1,2,3a&b). Why was the fusion not used as presented in Fig2?
- Table 3a and 3b are similar to the bottom two rows of Table 1 and 2 respectively. No new information is presented. I recommend to reference table 1 & 2 and use the space of Tables 3a&b to further explain the image-wise aggregation or the experimental setup to clarify why  temporal-attention fusion was not used in all experiments.
- Both configurations in Table 3c outperform the results of Table 3a&b. How does the ablation pretraining and fine-tuning change when applying any of the video aggregation methods?
- The paper cites many non-peer-reviewed publication on arXiv. Check if peer-reviewed versions are available (esp lines 539 & 545)
- Some references (lines 608, 623 & 630) are missing details on the journal / publisher

---

> ### Author Response · Authors · 2024-11-19
>
> We greatly appreciate your thorough review and constructive feedback. Your comments and suggestions were instrumental in improving both the clarity and quality of our paper.  We have carefully read through them and made the following changes in our revised paper:
>
> - We have changed Endovis18 in the experiment section to RSS (L319).
> - We have added a reference to Fig. 1 (L93).
> - We have updated Fig. 2 to improve the text decoder part.
> - We changed the paragraph starter from discussion to implications (L235).
> - Updated the references if applicapable (L608, L623, L630, L539, L535).
> - We have removed replicated sub-tables in Table 4 and added an ablation study for supervised video recognition.
> - We have updated the description to the image-based method for video recognition. (L431).
>
> We address the other questions below:
>
> > Line 184ff: N and T both represent a number of features, the final embedding size is NxTxD, its unclear what the difference between N&T is and why there are two numbers of frames.
>
> **R**: N refers to the number of frames sampled from the video, while T represents the number of image tokens extracted from each frame. The embeddings of one frame are TxD, and the embeddings for N frames are NxTxD. We have clarified this in the revised version.
>
> > Line 260: How are the generated tags split into the pretraining and fin-tuning set?
>
> **R**: Response The pretraining dataset consists of labels generated from our label engine, as described in Section 3, while the fine-tuning dataset includes VLM-annotated tags. For supervised experiments, the fine-tuning dataset also incorporates the training data from CholecT50.
>
> > Table1: EndoVis18 (RSS) is listed with 997 images. The dataset consists of 19 snippets with 300 frames each, why/how was it down sampled?
>
> **R**: EndoVis18 (RSS) has a public test split. We use the test split without any changes, which is publicly available [here](https://endovissub2018-roboticscenesegmentation.grand-challenge.org/Downloads/.). The test split includes four sequences with 249, 248, 248, and 248 labeled frames, totaling 977 images.
>
> > Table2: why aren't the values of P & R highlighted? Esp. for P the text references the models superiority, this should be visualized in the table.
>
> **R**: Highlighting precision (P) or recall (R) individually can be misleading without considering the balance between the two. For example, on the CLIP dataset, CLIP achieves a recall rate of 64.7%, but its precision is only 16.6%. In contrast, RASO achieves a recall rate of 43.3% with a significantly higher precision of 27.0%. Furthermore, by adjusting the threshold, RASO can achieve a recall rate of 64.8% with a precision of 21.8%. The inclusion of P and R in Table 2 aims to provide a more comprehensive understanding of model performance, rather than emphasizing one metric over the other. For more direct comparisons, metrics like F-beta and mAP, which aggregate both precision and recall, are more appropriate. We believe these metrics better reflect the overall performance and effectiveness of the models in question.
>
> > Why is the Fbeta with beta=0.5 instead of the more common F1 score used?
>
> **R**: The choice of F-beta with β=0.5 instead of the more common F1 score (β=1) reflects our effort to balance precision (P) and recall (R) more effectively. The β parameter adjusts the relative importance of P and R. While F1 gives equal weight to both metrics, we found that using β=1 often results in excessively high recall at the expense of precision for most methods. To address this imbalance, we chose β=0.5 to give slightly more weight to precision, which is critical for tasks requiring high specificity. For example, the table below shows the performance on the GraSP dataset, comparing results for β=0.5 and β=1.0 (F1 score):
>
> | Beta   | 0.5 ||  | 1.0 | |  |
> |---------|---------------|------------|--------|---------------|------------|--------|
> | Model   |  Precision (P) | Recall (R) | F-Beta | Precision (P) | Recall (R) | F-Beta |
> | CLIP    | 22.5          | 48.6       | 15.2   | 12.8          | 72.1       | 19.6   |
> | SurgVLP | 19.9          | 38.7       | 19     | 15.1          | 59.3       | 22.5   |
> | RASO    | 25.2          | 52         | 24.1   | 20            | 69         | 27.6   |
>
>
>
> > Line 417ff: Figure2 introduces the RASO architecture to include a temporal-attention fusion, but the video recognition paragraph compares the video aggregation against a frame-wise approach? How does the frame-wise method work? Which features are aggregated?
>
> **R**: For the frame-wise method, we run inference for each frame independently, and the predicted labels from all frames are aggregated using a union operation. Thus, we don't aggregate the features here.

---

> > ### Author Response · Authors · 2024-11-19
> >
> > > From the video recognition paragraph an Table3c it seems that the temporal-attention fusion was not used in the experiments of Table 1 and 2, as both configuration in Table3c differ from the earlier presented results (Table 1,2,3a&b). Why was the fusion not used as presented in Fig2?
> >
> > **R**: RASO processes the image and video inputs differently. The temporal fusion layer is only used when it receives video inputs. In Tables 1 and 2, we conducted image recognition experiments, because previous works in surgical object recognition, which we compare against in Table 2, are primarily image-based models. Temporal-attention fusion was evaluated separately for video-based recognition tasks in Figure 4 and Table 4.
> >
> > > Table 3a and 3b are similar to the bottom two rows of Table 1 and 2 respectively. No new information is presented. I recommend to reference table 1 & 2 and use the space of Tables 3a&b to further explain the image-wise aggregation or the experimental setup to clarify why temporal-attention fusion was not used in all experiments.
> >
> > **R**: We have improved the paper based on the suggestions. See L427 and Table 4.
> >
> > > Both configurations in Table 3c outperform the results of Table 3a&b. How does the ablation pretraining and fine-tuning change when applying any of the video aggregation methods?
> >
> > **R**: We have added an additional ablation study in Table 4 (L437) that studies the effectiveness of Pretraining for supervised video recognition tasks. For supervised experiments, fine-tuning is applied to all experiments. The reason that both configurations in Table 3c outperform the results of Table 3a&b is the differences between the image recognition labels and video recognition labels. For video recognition, the labels are generated by computing the union of objects appearing across all frames of a video. As a result, the video recognition results in Table 3c are not directly comparable to the image-based results in Tables 3a and 3b, as they are evaluated on different label sets and task definitions.

---

> > ### Comment · Reviewer_A4T2 · 2024-11-24
> >
> > Thanks for clarifying the open questions, all my comments were addressed, one minor thing:
> > Please replace the arXiv references:
> > Line 539 is available via ICRL
> > Line 546 is available via ICCV

---

> > > ### Author Response · Authors · 2024-11-26
> > >
> > > Thank you for the careful read and suggestions. We have updated the references in the latest revision.

---

### Official Review · Reviewer_B5E4 · 2024-11-03

**Soundness:** 3
**Presentation:** 2
**Contribution:** 2
**Rating:** 8
**Confidence:** 3

**Summary:**

The paper introduces a recognize-any-object (RAM) foundation model for surgery. The work employs a weakly supervised approach by generating a weakly annotated dataset from web-surgical-like data sources. Data from surgical lectures is processed to generate captions that are further used to generate different levels of annotations, including actions and entities. From the methodological perspective, the model follows RAM-based architectures with an additional temporal attention fusion layer integrated to work on video information.

**Strengths:**

The paper incorporates weak annotations from multiple video sources and investigates the zero-shot generalizability of adapted recognize-anything models in the surgical environment. Generating annotations from surgical videos is a common limitation in the medical domain as it requires expert knowledge. Hence, an approach that allows leveraging this data is interesting from the surgical ML perspective.

**Weaknesses:**

The work presents a modification to a previous RAM model and a pipeline for data generation. Even though leveraging web-based data from unstructured medical sources can be relevant for medical machine-learning purposes, I am unsure about the significance of the contribution from the general machine-learning perspective.

**Questions:**

The work introduces a methodology to generate weak annotations, which can also make it possible to fine-tune existing foundation models with extensive surgical data. In this regard, what are the additional benefits of the introduced model compared with standard foundation models fine-tuned on the weakly annotated data?

It is interesting to see that in Table 4, average pooling performs better than temporal attention in the instrument recognition tasks. What would be the main insights regarding this result?

Figure 3 is written WshiperX, probably instead of WhisperX.

---

> ### Author Response · Authors · 2024-11-19
>
> Thank you for taking the time to review our paper and for providing valuable feedback. Your comments have helped us identify areas where further clarification and improvements were needed. We have addressed your concerns in detail below.
>
> > The work presents a modification to a previous RAM model and a pipeline for data generation. Even though leveraging web-based data from unstructured medical sources can be relevant for medical machine-learning purposes, I am unsure about the significance of the contribution from the general machine-learning perspective.
>
> **R**:  Our work makes significant contributions to general machine learning by addressing both a fundamental data challenge and a structural limitation in current recognition models. First, our approach tackles a key ML challenge—constructing datasets in domains where manual annotation is costly or requires expertise. By leveraging domain-specific cues, such as transcripts and voiceovers, our pipeline generates accurate image-label pairs with minimal manual intervention. This general problem exists across many fields, making our pipeline broadly applicable. For instance, beyond surgery, it can be adapted to instructional videos in general medicine, chemistry, or mechanical engineering, enabling tasks like automated knowledge extraction or content summarization. Second, RASO is a novel video recognition structure that extends existing image-based models to effectively process videos. We introduce a temporal-fusion layer to capture temporal dependencies between frames, enabling efficient video recognition.
>
> > The work introduces a methodology to generate weak annotations, which can also make it possible to fine-tune existing foundation models with extensive surgical data. What are the additional benefits of the introduced model compared with standard foundation models fine-tuned on the weakly annotated data?
>
> **R**: Compared to traditional foundation models like CLIP and surgical domain-specific models such as SurgVLP, RASO addresses a key limitation in the reliance on image-text pairs for training. These models often suffer from noise introduced by irrelevant textual content when using weakly supervised data, which hampers their ability to learn surgical-specific visual concepts. RASO mitigates this issue through a joint training approach that integrates both image-tag and image-text pairs, preserving the richness of image-text information while leveraging the precision of image-tag pairs to improve recognition of visual concepts. As shown in Table 2, RASO outperforms SurgVLP with mAP improvements of 2.9, 4.5, 10.6, and 7.2 across four benchmarks, demonstrating its superiority in surgical object recognition tasks.
>
> > It is interesting to see that in Table 4, average pooling performs better than temporal attention in the instrument recognition tasks. What would be the main insights regarding this result?
>
> **R**: In tasks involving the recognition of static tags, such as surgical instruments or anatomical structures, the goal is to identify the union of static tags appearing across frames. Since the order of video frames does not affect the result for these static tags, methods like average pooling that disregard temporal information can still perform effectively. However, for recognizing surgical actions, temporal information is crucial, as these actions depend on the sequential dynamics between frames. While Table 4 shows that average pooling outperforms temporal fusion by 1.4 mAP in instrument recognition tasks, the temporal attention mechanism brings a significant 6.0 mAP improvement in action recognition tasks, highlighting its importance for tasks that require capturing temporal dependencies.
>
> > Figure 3 is written WshiperX, probably instead of WhisperX.
>
> **R**: Thank you for pointing this out. We have corrected the spelling in Figure 3.

---

> > ### Comment · Reviewer_B5E4 · 2024-11-26
> > **Reply to Author's feedback.**
> >
> > Thank you for the additional details and for addressing the questions. I have no additional comments. I have decided to change my rating to accept.

---

> > > ### Author Response · Authors · 2024-11-26
> > >
> > > Thank you for your thoughtful feedback and for reconsidering our work. We appreciate your support.

---

### Official Review · Reviewer_ZNLc · 2024-11-04

**Soundness:** 3
**Presentation:** 4
**Contribution:** 3
**Rating:** 8
**Confidence:** 4

**Summary:**

The authors introduced a foundation model aimed at recognizing surgical objects in closed and open set environments. It also adapts to learn from images and videos. They introduce a weakly supervised learning framework to generate tag-image-text pairs from surgical lecture videos. This allowed them to create an interesting dataset that could be helpful for researchers. Experimental results demonstrate that the proposed method outperforms other SOTA methods in zero-shot scenarios and in supervised surgical action recognition tasks.

**Strengths:**

- The paper is well presented and easy to follow.
- Although the technical contribution is limited, the efforts and contributions in the paper are great and are likely to help the research community in the field of surgical tool detection, segmentation and analysis.
- The evaluation is good and ablation studies reflect how the method could work in different settings.

**Weaknesses:**

1- The main weakness of the paper from a technical perspective is the limited technical contribution. The methodology is heavily based on RAM (Zhang 2024) with an additional temporal attention layer.

2- It is not clear in Figure 2 how the embeddings from I and Vs are processed in the Tag Decoder. How does the network balance the importance of information coming from images or videos?

3- It was not clear why pre-training and fine-tuning were performed for a small number of epochs. Does the loss saturate when fine-tuning for 4 epochs only?

4- VLM automated annotation: it is not clear the effect of the ability of GPT-4o in generating accurate annotation. An assessment of this step is needed.

**Questions:**

To relate to the 4 points in the weakness section

1- Explain or confirm the main technical differences with RAM

2- Provide more details on how the network balances the importance of image and video embeddings

3- Answer point 3 in the weaknesses

4- Elaborate on 4 in the weaknesses

---

> ### Author Response · Authors · 2024-11-19
>
> We sincerely thank you for your thoughtful feedback and detailed comments. We appreciate your recognition of our efforts in presenting a well-structured paper and conducting comprehensive evaluations. Below, we address your concerns and clarify the points you raised.
>
> > It is not clear in Figure 2 how the embeddings from I and Vs are processed in the Tag Decoder. How does the network balance the importance of information coming from images or videos?
>
> **R**: We appreciate your observation. For image inputs, embeddings are generated using the Image Encoder and passed along with tag embeddings through the Tag Decoder to produce tag predictions. For video inputs, we uniformly sample 15 frames from the video and process each frame through the same Image Encoder to generate frame-level embeddings. These embeddings are then combined using a temporal-fusion layer, which captures temporal dependencies across frames, to form a unified video embedding. This aggregated video embedding, together with tag embeddings, is then passed through the Tag Decoder to generate predictions. We have updated the Figure 2 by adding a concat operation to explain how frame tokens are aggregated more clearly.
> Our network does not explicitly "balance" the importance of images and videos; instead, it relies on the shared Image Encoder for consistent frame-wise feature extraction and the temporal-fusion layer for capturing video-specific temporal information. This design ensures that both image and video inputs are projected into a **unified visual representation space**.
>
> > It was not clear why pre-training and fine-tuning were performed for a small number of epochs. Does the loss saturate when fine-tuning for 4 epochs only?
>
> **R**: Our choice of a small number of pretraining and fine-tuning epochs stems from the scale of our weakly-supervised dataset and the pre-initialization from RAM weights. The pretraining dataset includes approximately 900K images, allowing RASO to learn effectively within 10 epochs. For fine-tuning, we find 4 epochs to be a sweet spot, balancing performance and training time. The fine-tuning dataset is much smaller than the pretraining dataset, which helps mitigate the risk of overfitting during fine-tuning.
>
> > VLM automated annotation: it is not clear the effect of the ability of GPT-4o in generating accurate annotation. An assessment of this step is needed.
>
> **R**: We recognize the importance of evaluating the quality of GPT-4o-generated annotations.  The intuition behind using GPT-4o for annotation is that we observed GPT-4o tends to use common tags with reasonable accuracy, which complements the broader tag coverage of the pretraining dataset. This synergy enhances the model's performance on frequently occurring tags. To this end, we conducted an assessment by randomly sampling 1,000 annotated images and assigning them to seven clinicians for expert tag annotations. The agreement rate, defined as the percentage of tags matched between GPT-4o annotations and clinician annotations, was calculated for 20 frequent tags. The results are as follows:
>
> | Tag                | Mediastinum | Spleen | Peritoneum | Abdominal Wall | Stapler | Intestine | Pelvic Cavity | Needle Driver | Ligasure | Lung  | Gallbladder | Scissors | Harmonic Scalpel | Trocars | Hernia Sac | Choledochal Cyst | Adipose Tissue | Suture | Laparoscopic Grasper | Skin  |
> |---------------------|-------------|--------|------------|----------------|---------|-----------|---------------|---------------|----------|-------|-------------|----------|------------------|---------|------------|------------------|----------------|--------|----------------------|-------|
> | Agreement Rate      | 0.35        | 0.32   | 0.32       | 0.31           | 0.29    | 0.29      | 0.27          | 0.27          | 0.25     | 0.25  | 0.24        | 0.22     | 0.21             | 0.21    | 0.21       | 0.20             | 0.20           | 0.19   | 0.19                 | 0.19  |
>
> These results indicate that GPT-4o provides annotations for common surgical objects with moderate accuracy, offering a reliable foundation for weak supervision.
> >  Explain or confirm the main technical differences with RAM
>
> **R**: The primary technical difference is that RASO extends RAM's capabilities from image-only tasks to video recognition with improved accuracy and effiency. This is achieved through the introduction of a temporal-fusion layer that captures temporal information across video frames, enabling more robust video-based recognition. Additionally, RASO introduces a general and scalable data generation pipeline to construct a large-scale weakly-supervised surgical dataset from surgical lecture videos, significantly enhancing its domain-specific applicability.

---

> > ### Comment · Reviewer_ZNLc · 2024-11-26
> > **Response to author comments**
> >
> > Thanks for the response. I think mentioning the evaluation of GPT4o would be good. I have no more comments.

---

> > > ### Author Response · Authors · 2024-11-26
> > >
> > > Thank you for your feedback and recognition of our work. In the latest revision, we have included an evaluation of GPT-4o in the appendix.

---

### Author Response · Authors · 2024-11-19

We sincerely thank all reviewers for their thoughtful feedback and constructive comments. Your insights have been invaluable in improving both the clarity and quality of our work. **We have submitted a revised version of the paper**, where all updates and modifications are highlighted in the **blue font** for ease of review. Below, we provide a summary of the key improvements and responses made in the revised paper. Detailed individual responses to each reviewer’s comments can be found below.

 **Major Improvements**

1. Figure Updates:
  - Updated Figure 2 to clarify how image embeddings are aggregated for video input, and provide a better illustration of the text decoder for transcript reconstruction joint training.
  - Added a word cloud in Figure 3 to provide an overview of tag frequency, offering a preliminary observation of the dataset’s entity keyword distribution.
2. Revised Text and Tables:
 - Renamed EndoVis18 to RSS in the experiment section and updated related descriptions.
 - Improved Table 4 by cleaning the redundant content and adding an ablation study for supervised video recognition.
 - Clarified the use of temporal-attention fusion for video recognition versus frame-wise aggregation for image-based video recognition baseline.

Thank you again for your valuable time and effort. We hope the revisions and clarifications address your concerns.

---

### Public Comment · ~Ka_Young_Kim1 · 2025-12-29
**Clarification on CholecT50**

Thank you very much for your interesting work.

I especially enjoyed reading the experiments on CholecT50, as many existing surgical VLM studies have mainly focused on phase recognition tasks.

As far as I understand, CholecT50 provides several different dataset splits (e.g., as described in https://arxiv.org/pdf/2204.05235). I was wondering if you could kindly clarify which split was used in your experiments.

I was also curious about how frames without any annotated triplets were handled in your experimental setup.

Finally, I would be grateful if you could share a bit more insight into how the threshold was chosen for the zero-shot evaluation on CholecT50. For example, was it tuned on a validation set, fixed empirically, or selected based on some principled criterion?

---

> ### Public Comment · ~Jiajie_Li2 · 2025-12-29
> **Clarification on CholecT50 Experimental Setup**
>
> Thank you very much for your interest in our work and for your thoughtful questions.
>
> Regarding the dataset split, we use the train/validation/test split from the Rendezvous publication[1], as reported in Table 2 of the paper you referenced, and apply it consistently in our experiments.
>
> For frames without annotated triplets, we do not perform any special filtering or handling. These frames are kept in the evaluation, which may lead to slightly lower results compared to setups that exclude empty-annotation frames. The same protocol is applied to all models and benchmarks to ensure fair comparison.
>
> For zero-shot evaluation on CholecT50, we select thresholds per class and per model by maximizing the F-β score on the test set. We then report the optimal F-β together with the corresponding precision and recall.
>
> We hope this clarifies your questions, and we appreciate your careful reading and constructive feedback.
>
> [1] Nwoye, Chinedu Innocent, et al. "Rendezvous: Attention mechanisms for the recognition of surgical action triplets in endoscopic videos." Medical Image Analysis 78 (2022): 102433.

---

### Meta-Review · Area_Chair_wFfa · 2024-12-10

**Metareview:**

The submission presents a foundation model aimed at segmenting surgical objects in closed and open set environments.  The reviewers were unanimous that the submission should be accepted.  The significance is primarily in providing a seemingly effective solution to automated surgical video annotation with minimal labeling, leveraging an unannotated dataset.  Effectiveness is shown in zero-shot settings, as well as in supervised surgical action recognition tasks.  On the balance, the submission provides an interesting solution to a medical application, which should find its audience in the ICLR and surgical video analysis communities.

**Additional Comments On Reviewer Discussion:**

The authors were very responsive, and there was some degree of interaction from the reviewers.  The authors were comprehensive in their responses and the evaluation of GPTo is a concrete outcome from this process that was appreciated by the requesting reviewer.

---

### Decision · Program_Chairs · 2025-01-22

Accept (Spotlight)